# Microstructure and Dry/Wet Tribological Behaviors of 1% Cu-Alloyed Austempered Ductile Iron

**DOI:** 10.3390/ma16062284

**Published:** 2023-03-12

**Authors:** Cheng-Hsun Hsu, Chun-Yin Lin, Wei-Shih You

**Affiliations:** Department of Mechanical and Materials Engineering, Tatung University, Taipei City 10451, Taiwan; rupert475@gmail.com (C.-Y.L.); mi155438@gmail.com (W.-S.Y.)

**Keywords:** copper alloying, austempered ductile iron, microstructure, hardness, wear resistance

## Abstract

In this study, different austempering conditions were applied to 1 wt.% Cu-alloyed ductile iron to produce various austempered ductile irons (ADIs). The study aimed to explore the variations in microstructure, hardness, and dry/wet wear behaviors of the ADIs. The experimental results indicated that the microstructure of the 300 °C–ADI has denser needle-like ausferrite, lower retained austenite content, and higher carbon content in austenite compared with the 360 °C–ADI. As the austempering time increased, the retained austenite content decreased, while the carbon content of austenite increased. Regardless of dry or wet abrasive behavior, the wear resistance of the ADIs was significantly superior to that of the as-cast material. The ADI obtained at 300 °C for 10 h demonstrated the best wear resistance performance.

## 1. Introduction

Ductile iron (DI), also known as nodular graphite cast iron, is a type of cast iron with a graphite-rich microstructure. It was first developed by the British Cast Iron Research Association (BCIRA) in 1948. Because of its excellent castability, machinability, and mechanical properties, as well as its low cost, DI has been widely used in various industrial products, including pipes, machine tools, and automotive parts [1,2].

In general, DI can be modified by alloying and heat treatment to obtain the desired mechanical properties. For example, elements such as Cr, Mo, Sn, and V are strong pearlite formers that can increase hardness and strength [1,2]. Copper is a common alloying addition to cast iron as it is an austenite stabilizer, which can delay the start of the transformation to pearlite. The overall effect of adding up to 0.35% copper to cast iron is an increase in tensile strength, hardness, and corrosion resistance [3,4].

Furthermore, heat treatment is a crucial technology for most cast irons used in practical applications [5]. For instance, the austempering treatment of DI can significantly improve its mechanical properties, resulting in austempered ductile iron (ADI) with improved hardness, toughness, and fatigue properties [6,7,8,9,10,11,12,13,14]. Because of these advantages, ADI is often used in gears, suspension brackets, rails, brake blocks, and other materials requiring high contact stress. Therefore, the wear resistance of ADI is crucial in engineering applications. While some studies have investigated the effects of surface hardening and alloying on the wear resistance of ADI [15,16,17,18,19], information regarding the austempering effect remains insufficient [20,21,22].

The motivation of this work was to study copper (1 wt.% Cu)-alloyed DI as the primary material. The iron was austempered under different conditions of temperature and time to obtain various ADIs. The microstructure, hardness, and dry/wet wear behavior of the ADIs were observed and measured. This is because ADI is often used in environments with dry wear behavior, such as automotive gear components. The wet wear test, in particular, is an innovative aspect of this study as it simulates the performance of ADI in environments with a seawater atmosphere, such as offshore wind turbine components. The influence of austempering parameters on the microstructure, hardness, and dry/wet wear resistance of DI was investigated. The results of this study can provide a valuable reference for relevant research and practical applications.

## 2. Experimental Procedure

### 2.1. Material and Specimen Preparation

In this study, DI with 1 wt.% Cu addition was used as the experimental material. The molten iron was cast into a Y-block green sand mold to obtain the casting. Specimens were cut from the rectangular part below the casting and machined into disc-type specimens with a diameter of 20 mm and a thickness of 3 mm, which were required for the wear tests. Three specimens were prepared for each experimental condition. The chemical composition of the DI was analyzed using glow discharge optical spectroscopy (GDOS), and the results are shown in Table 1.

### 2.2. Austempering Treatment

The DI underwent austempering treatment, as shown in Figure 1. To improve the mechanical properties of ADI based on previous studies [12,13,14], the following temperature processing method was adopted in this study. The specimens were preheated at 550 °C for 15 min in box-type furnaces (SYA-S1, 600 kg), austenitized at 910 °C for 1.5 h, and then quenched in a salt bath for isothermal treatment. Specifically, the specimens were separately austempered at 300 °C and 360 °C for different durations (1, 5, and 10 h) to produce various ADIs. The specimens were coded as 300–1 h, 300–5 h, 300–10 h, 360–1 h, 360–5 h, and 360–10 h, respectively.

### 2.3. Metallographic Test

The specimens underwent sequential grinding with different grades of sandpaper (80#, 240#, 400#, 800#, and 1200#) followed by polishing with 0.3 μm alumina powder. The microstructures were observed using an optical microscope (OM) after chemical etching with a 5 wt.% Nital solution (95 mL C_2_H_5_OH + 5 mL HNO_3_).

### 2.4. XRD Analysis and Hardness Test

X-ray diffraction (XRD) tests were performed on the ADI specimens using a copper target X-ray diffractometer (Bruker AXS GmbH D2 PHASERX) with a voltage of 30 kV, a current of 30 mA, a scanning speed of 2.5 degrees/min, and a diffraction range of 40° to 100°. The volume fraction of retained austenite (Vγ) in the microstructure was calculated using Equation (1) as shown below, based on previous studies [12,13]:Vγ% = (Iγ/Rγ)/[(Iγ/Rγ) + (Iα/Rα)] × 100% (1)
where Iγ and Iα are the integrated intensity for austenite and ferrite, respectively, and Rγ and Rα are the theoretical relative intensity for austenite and ferrite, respectively [23]. Additionally, the carbon content of austenite (Cγ) was calculated using Equation (2), which describes the relationship between Cγ and aγ:Cγ = (aγ − 3.548)/0.44 (2)
where aγ is the lattice parameter of austenite determined by X-ray measurement [24].

For hardness testing, a Rockwell hardness tester (Matsuzawa Seiki MARK-M2) was used to measure the hardness (HRC) values of the as-cast DI and obtained ADIs at room temperature with a load of 150 kg. Each specimen was tested five times to obtain an average hardness value.

### 2.5. Dry and Wet Wear Tests

Two types of wear tests, i.e., dry and wet, were conducted in this study. The wet wear tests aimed to simulate the corrosive wear behavior of the specimens in seawater, as would be encountered in offshore wind turbine assembly applications. A 3.5 wt% sodium chloride solution was used as the testing medium. Another test is the dry wear test that simulates the application of automotive gear components. A multi-functional abrasion tester (CSM model CW73RL) was employed to conduct the ball-on-disc wear tests of the specimens against a grinding load of 2 N, turning radius of 5 mm, and a total abrasion distance of 500 m. The abraded ball was made of WC-6%Co with a diameter of 6 mm. The testing environment for dry abrasion tests was 65% humidity at room temperature. Before the testing, the specimens were cleaned by ultrasonic shock for 10 min in an alcohol solution. The data obtained from each wear test were used to draw a diagram of the relationship between wear distance and friction coefficient. Before and after the test, the specimen’s weight was measured using a precision electronic scale (±×10^−4^ g) to calculate the wear rate. The surface abrasion tracks of the specimens were observed using OM and SEM.

## 3. Results and Discussion

### 3.1. Microstructure Observation

Figure 2 depicts the microstructure of the 1 wt.% Cu-alloyed DI. The matrix has a typical bull’s-eye structure consisting of a ferrite and pearlite mixture, along with the presence of nodular graphite. After austempering with varying temperature and time, the ADIs were obtained, the microstructures of which are shown in Figure 3. Figure 3a–c represent the microstructure of ADIs austempered at 300 °C for 1, 5, and 10 h, respectively. At 300 °C austempering temperature, the nucleation force was higher, and the growth rate was slower for ausferrite formation. This is because of the large degree of undercooling and the diffusion-dependent growth rate at lower temperature [5]. Consequently, a dense, needle-like ausferrite structure formed with a relatively small amount of retained austenite (white areas). On the other hand, at a higher austempering temperature of 360 °C, the nucleation rate of ausferrite was slower, and carbon atoms diffused faster, resulting in a sparse and feathery appearance of ausferrite morphology, and more retained austenite in the microstructure (white areas) as shown in Figure 3d–f. Overall, the effect of austempering temperature on microstructural morphology seemed to be more prominent than that of holding time, implying that the ADI microstructure is primarily determined by the degree of undercooling.

Figure 4 shows the XRD patterns of the ADIs obtained in this study. Upon comparison, it was found that all ADIs had a similar composition which primarily comprised ausferrite and retained austenite phases. This result is consistent with the microstructures observed in Figure 3 and complies with the standard specification for ADI [25]. Several studies [26,27,28] have suggested that the retained austenite content in the microstructure can affect the mechanical properties and corrosion resistance of ADI. For example, a higher amount of retained austenite is associated with better toughness and corrosion resistance. In this study, the retained austenite content and the carbon content in austenite were obtained and compared based on X-ray measurements, as shown in Figure 5.

Figure 5a,b respectively show the effects of austempering parameters on the retained austenite content and carbon content in retained austenite. The results indicated that a higher retained austenite content was obtained at an austempering temperature of 360 °C. Additionally, the retained austenite content decreased with an increase in austempering time. The cause for the tendency of the change in retained austenite content is discussed in the previous section. The copper-alloyed ADIs in this study had more retained austenite than the unalloyed ones in past studies [28], which can be attributed to the stabilizing effect of copper on the austenite, as mentioned earlier [3,4]. In terms of carbon content in austenite, higher carbon content was observed at a lower austempering temperature of 300 °C, where the 300–10 h specimen had the highest carbon content (1.93 wt.%C) in austenite. This is because the diffusion rate of carbon is slower at a lower temperature, resulting in a higher carbon content in the retained austenite. Similarly, a higher solid-soluted carbon content existed in the retained austenite as the austempering time increased.

### 3.2. Austempering Effect on Hardness of ADI

The average hardness values of the various ADIs were obtained through hardness tests. Figure 6 shows a comparison of these hardness values for as-cast DI and the ADIs. It can be observed that the hardness values of the ADIs (31–45 HRC) were significantly higher those that of the as-cast DI (16 HRC). Among the ADIs, the hardness of the ADI austempered at 300 °C was higher than that at 360 °C. This is because the former has a dense needle-like ausferritic microstructure, lower retained austenite content, and higher carbon content in austenite. Furthermore, extending the austempering duration of the ADI specimens can decrease the amount of retained austenite and result in a higher hardness level. Notably, the 300–10 h ADI exhibited the maximum hardness value of 45 HRC. The hardness of the ADIs obtained in the study is compliant with the level of ASTM specification [25].

### 3.3. Wear Behavior Analysis

In this study, two types of wear tests, dry and wet, were conducted, and the results were compared and discussed as follows. In the case of the dry wear test, the friction coefficient and wear distance relationship for each specimen after testing were obtained, as shown in Figure 7. Comparing these friction coefficient curves, it can be seen that all the ADIs had a lower friction coefficient than the as-cast DI (0.1–0.4 vs. 0.5). Moreover, the friction coefficient of the 300 °C–ADI was lower than that of the 360 °C–ADI. This result is in accordance with the related literature [10] and mainly depends on microstructure. That is, the ADI austempered at a lower temperature had a lower friction coefficient and showed good tribological behavior due to its denser microstructure. In particular, the 300–10 h ADI had a minimum friction coefficient of about 0.1.

Figure 8 shows a comparison of wear rates among the as-cast DI and the various ADIs after dry wear testing. It is evident that the wear rate of ADIs was much lower than that of DI. The mechanism behind the increased wear resistance of ADI is inferred to be due to the presence of retained austenite in the microstructure, which generates martensite transformation under loading [18,29]. Additionally, it shows that both the friction coefficient and wear rate of the 300 °C–ADIs were lower than those of the 360 °C–ADIs. The 300–10 h ADI had the lowest friction coefficient and the lowest wear rate, demonstrating the best dry wear resistance performance. This result is consistent with the literature [30], which indicates a direct correlation between wear rate and hardness.

Figure 9 presents the abraded surface morphology of the specimens after dry wear testing. The results indicate that the width of the abrasion track decreased with an increase in hardness. The narrowest worn track was observed for the 300–10 h specimen, which had the lowest friction coefficient and the highest hardness, as indicated by the red dotted line in Figure 9d. This suggests that the dry abrasion behavior of ADI mainly depends on its hardness. Additionally, it was observed that the wear behavior of all the as-cast DI and ADIs specimens demonstrated a mixture of adhesive and abrasive wear mechanisms. However, compared with the ADI specimens, the softer as-cast DI exhibited more pronounced adhesive wear behavior, resulting in the widest worn track, as shown by the red dotted line in Figure 9a. The other ADI specimens (Figure 9b,c,e–g) had narrower worn tracks than as-cast DI, while being wider than the 300–10 h specimen. This observation regarding the wear appearance is indicative of the 300–10 h ADI specimen exhibiting superior wear resistance.

In the wet wear test, a 3.5 wt.% NaCl solution was used to simulate the corrosive wear behavior of the irons in seawater. Figure 10 and Figure 11 compare the friction coefficient and wear rate of all specimens after testing, respectively. Similarly, it was found that ADIs had lower friction coefficients and wear rates than as-cast DI. The abrasion mechanism is as described above, and wet abrasion was more severe because of the contact effect of the corrosive medium. Furthermore, when comparing the ADIs, the results showed that the 300–10 h ADI had the lowest friction coefficient and wear rate, indicating the best corrosive wear resistance.

Figure 12 displays the SEM surface patterns of the specimens after the wet corrosive abrasion tests. Besides the abrasion tracks, a few localized areas of graphite corrosion [28] can be observed. Notably, the as-cast DI specimen exhibited chloride (FeCl_3_) corrosion products around graphite (as indicated by the white dots seen in Figure 12a). In contrast, the corrosion phenomenon of the abrasion tracks of the ADI specimens appeared to be relatively minor, as shown in Figure 12b–g. More specifically, the wear trace on the 300–10 h ADI specimen was notably narrower and shallower, suggesting that it possessed the highest level of resistant corrosive wear performance among all specimens (Figure 12d).

Furthermore, when comparing the abrasion behavior of ADI between wet wear testing and dry wear testing (Figure 13), the friction coefficient was lower in the wet case because of the lubrication of NaCl solution, but the wear rate was higher because of both the damage effects of abrasion and corrosion. This is supported by the nearly identical wear rate trend of the 300–ADI in wet wear testing and the 360–ADI in dry wear testing. Regardless of dry wear testing or wet corrosive wear testing, the 300–10 h ADI exhibited the best abrasion resistance performance.

## 4. Conclusions

The austempering temperature mainly affected ADI’s microstructure. ADI austempered at 300 °C had a dense, needle-like ausferrite structure, while ADI austempered at 360 °C had a sparse and feathery appearance;ADI had much higher hardness than as-cast DI (31–45 vs. 16 HRC). The hardness of ADI was affected by austempering temperature, with 300 °C–ADI being harder than 360 °C–ADI. Among all ADI specimens, the 300–10 h ADI had the highest hardness value (45 HRC).Lower austempering temperature and longer austempering time can improve ADI’s wear resistance, whether in dry or wet corrosive wear tests. ADI treated with 300–10 h austempering showed excellent wear resistance performance.

## Figures and Tables

**Figure 1 materials-16-02284-f001:**
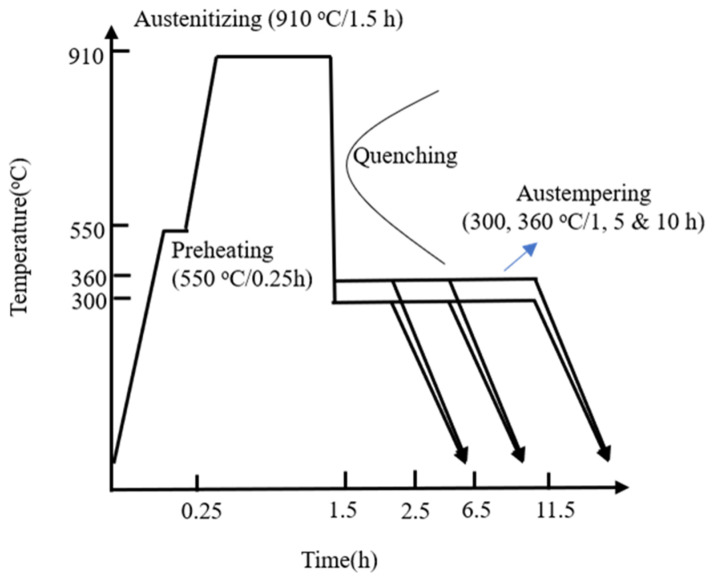
Flowchart of the austempering treatment in the study.

**Figure 2 materials-16-02284-f002:**
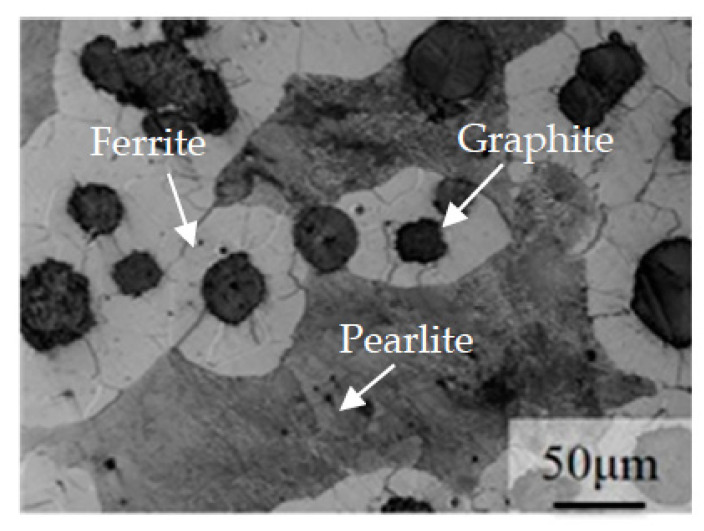
Microstructure of the ductile iron.

**Figure 3 materials-16-02284-f003:**
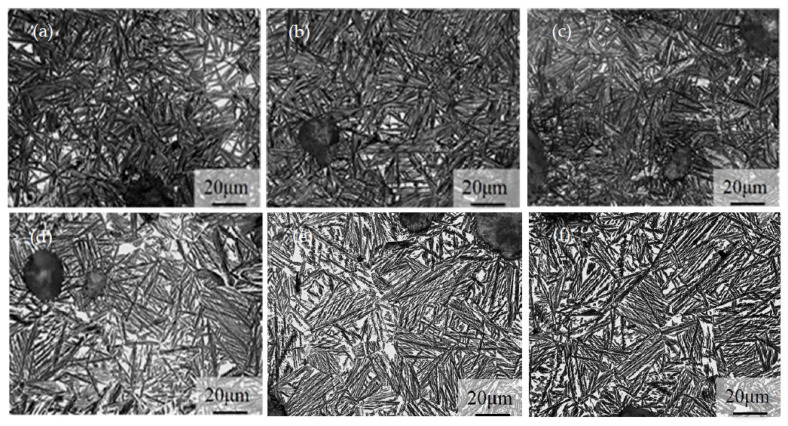
Microstructures of the ADIs: (**a**) 300–1 h, (**b**) 300–5 h, (**c**) 300–10 h, (**d**) 360–1 h, (**e**) 360–5 h, and (**f**) 360–10 h.

**Figure 4 materials-16-02284-f004:**
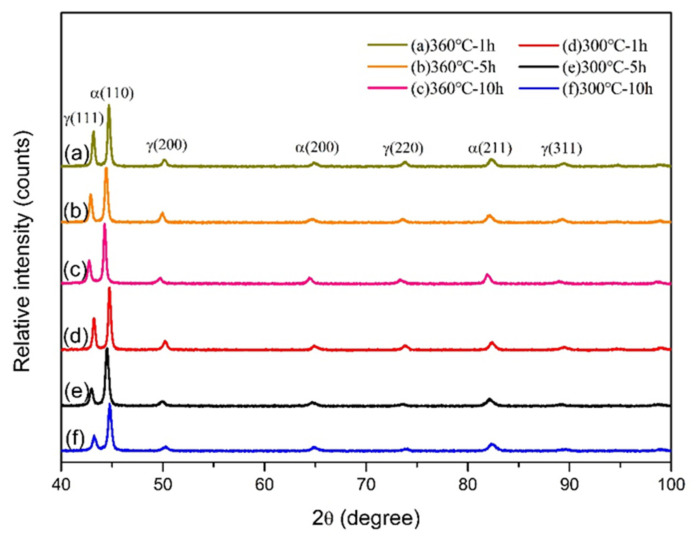
XRD patterns of the ADIs in the study.

**Figure 5 materials-16-02284-f005:**
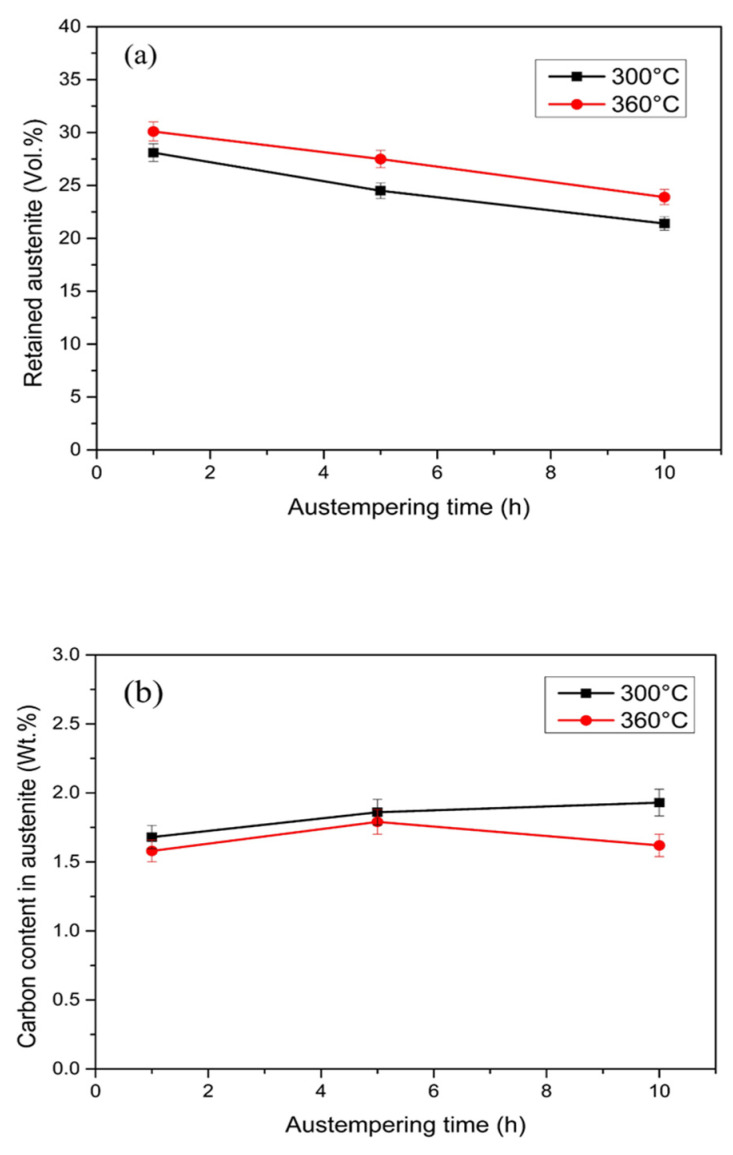
The effect of austempering parameters on (**a**) retained austenite content and (**b**) carbon content in austenite.

**Figure 6 materials-16-02284-f006:**
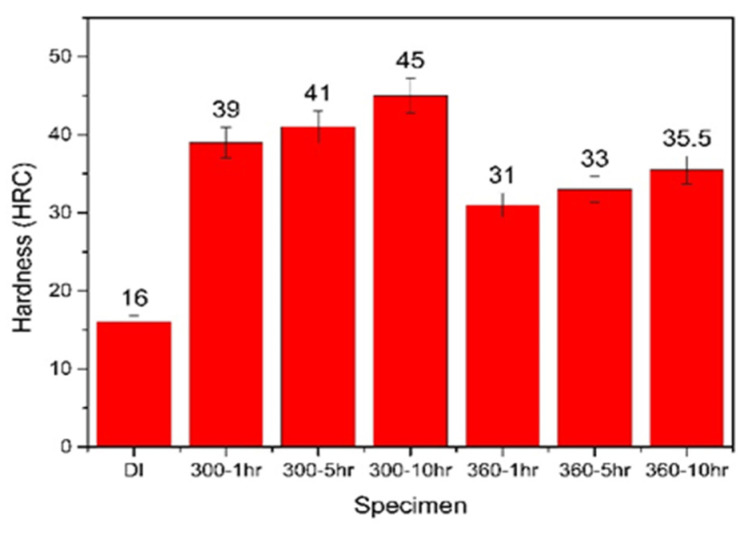
Comparison of the hardness values among as-cast DI and the various ADIs.

**Figure 7 materials-16-02284-f007:**
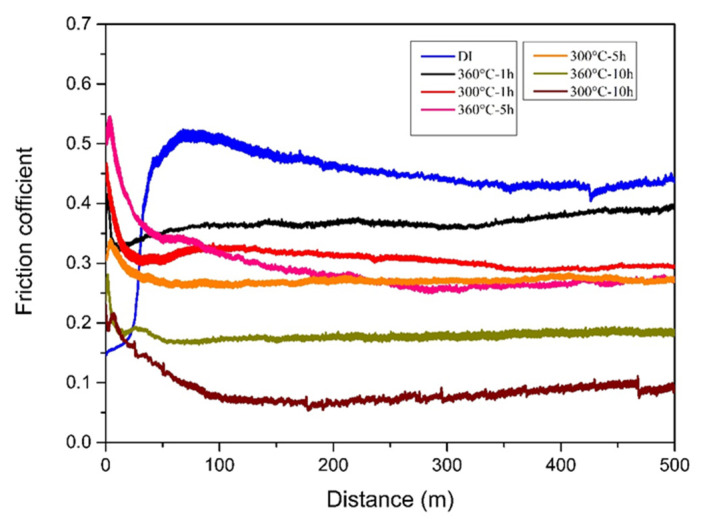
Comparison of friction coefficients among the specimens after dry wear testing.

**Figure 8 materials-16-02284-f008:**
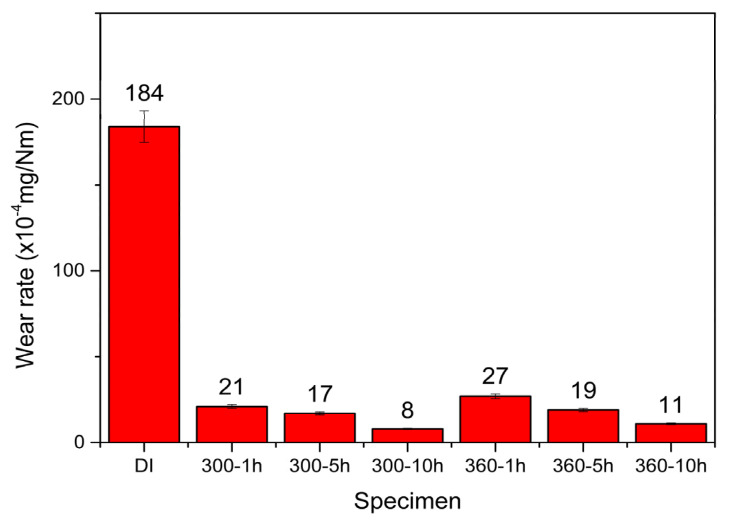
Comparison of wear rates among the specimens after dry wear testing.

**Figure 9 materials-16-02284-f009:**
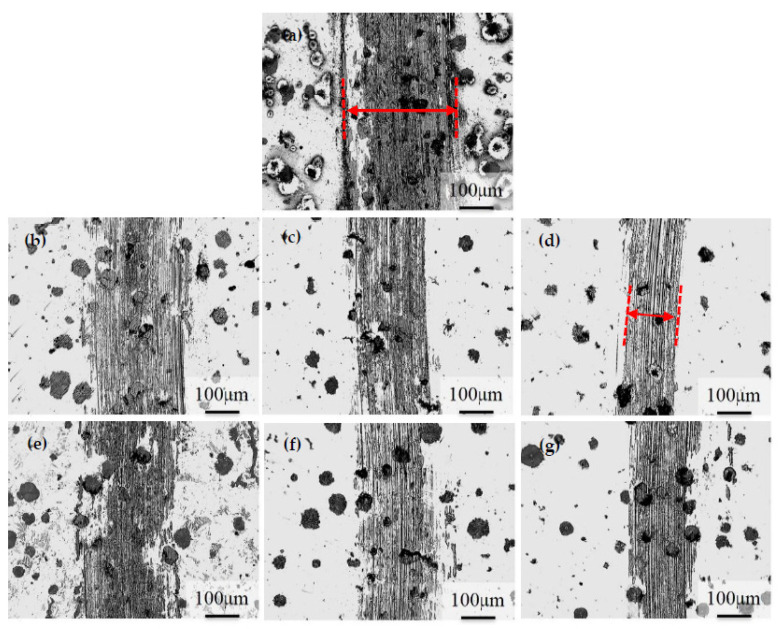
Surface worn tracks of the specimens after dry wear testing. (**a**) as-cast, (**b**) 300–1 h, (**c**) 300–5 h, (**d**) 300–10 h, (**e**) 360–1 h, (**f**) 360–5 h, and (**g**) 360–10 h.

**Figure 10 materials-16-02284-f010:**
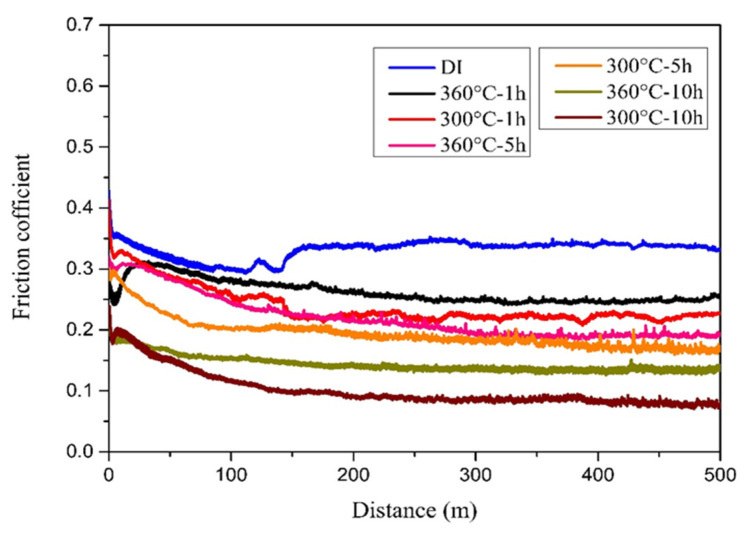
Comparison of friction coefficients among the specimens after wet wear testing.

**Figure 11 materials-16-02284-f011:**
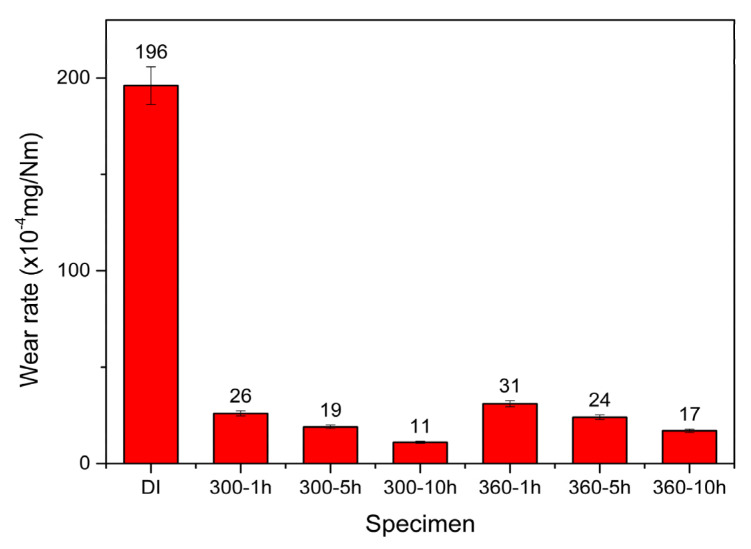
Comparison of wear rates among the specimens after wet wear testing.

**Figure 12 materials-16-02284-f012:**
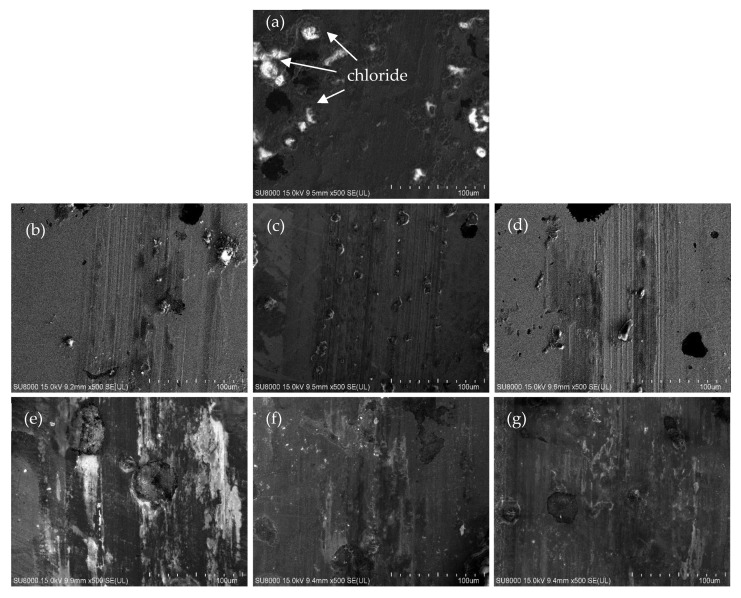
Surface worn tracks of the specimens after wet wear testing: (**a**) as-cast, (**b**) 300–1 h, (**c**) 300–5 h, (**d**) 300–10 h, (**e**) 360–1 h, (**f**) 360–5 h, and (**g**) 360–10 h.

**Figure 13 materials-16-02284-f013:**
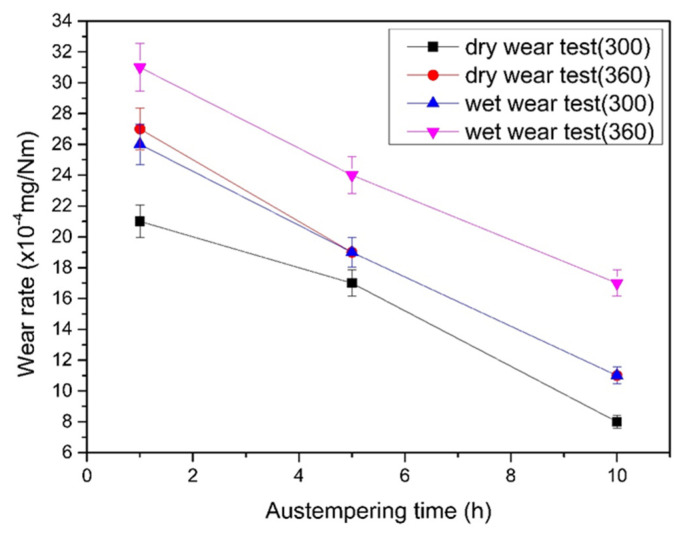
The effects of dry and wet wear types on the wear rate of the ADIs.

**Table 1 materials-16-02284-t001:** Chemical composition of the ductile iron in the study (wt.%).

C	Si	Mn	P	S	Cu	Fe.
3.42	1.93	0.24	0.05	0.02	1.01	Bal.

## Data Availability

The data presented in this study can be available upon request from the corresponding author.

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
