# Peer review of "Microstructure and Dry/Wet Tribological Behaviors of 1% Cu-Alloyed Austempered Ductile Iron"

_materials, 2023, doi:10.3390/ma16062284_

Round 1

Reviewer 1 Report

 Introduction

1- In the introduction there are words with hyperlinks to Wikipedia

2- The unit used in the sample dimensions is not the correct way to use it (mm3). It would be D20 mm x 3 mm

3- In figure 1 it is necessary to organize the text of the axes and improve the way of indicating the temperatures and times of each stage of the heat treatment shown.

2. Experimental Procedure

2.2. Austempering treatment

4-Line 66-67, page 2. The paragraph needs to be revised, as it is confusing: “First the specimens were austenitized at 910 oC for 1.5 h after preheated at 550 oC for 15 66 min, and then quenched into a salt bath for isothermal treatment.”

5- It is recommended to indicate the equipment (brand, capacity, etc.) used in heat treatments and other tests.

6- Line 66-68, page 2. What was the selection criteria for the temperatures and times of heat treatment?

7-Line 84 page 3. It is recommended to indicate the equation numbering after the word equation and the reference at the end of the sentence.

Results and discussion

8- There are some phenomenological statements that should be referenced (i.e. For example, the explanation of the microstructures in Figure 3).

9- Similarly to the previous comment, the explanation of the results shown on page 5 with respect to figure 5..

10-Lines 153-155, Page 5. What is the aforementioned increase in carbon due to?

11- How was the retained austenite determined?

12-Line 185-186, Page 6. Are the hardness levels with respect to the reported austempering temperatures in accordance with what is reported in the literature?

In general:

Some comments to take into account:

The authors are invited to address the effect of the segregation of the different alloying elements on the transformation kinetics, the microstructure and the mechanical properties of the ADI obtained under the indicated study conditions.

Basically, the article lacks an in-depth analysis of the phenomenological aspects of the transformation given by the austempering heat treatment carried out and from the perspective of the copper content as an alloying element.

Additionally, there is not an adequate discussion of the results since there is not enough contrast and with arguments based on the literature on ADIs, which is extensive.

The alloying elements are frequently added to favor the formation of specific phases in order to promote an improvement in the mechanical properties and/or austemperability; but due to micro-segregation, there are differences in the transformation kinetics between neighboring graphites. Micro-segregation in austempered nodular irons has a highly important effect in heat treatments, due to this, when processing the material, microstructural differences may occur throughout the matrix, therefore, its mechanical performance is will be significantly affected.

It would be interesting to present microstructures at different magnifications before and after etching before heat treatment.

Likewise, it would be interesting to show the microstructures resulting from the heat treatments at different magnifications. This would allow us to see, in addition to the formation of the phases expected by the heat treatment, how the graphite nodules are distributed.

Reviewer 2 Report

The paper begins by describing the historical development of the material. Subsequently, the authors highlight the effects of some alloying elements and the material application.

It is the reviewer's opinion that the introduction must explain the innovation of the study and respects the state of art.

row 65 Please put the figure in the test after their description

please evidence the repetition number of manufactured specimens for each test.

row 35  "ADI is often used in gears, suspension brackets, rails, brake blocks, and other materials requiring high contact stress"

please justify the seawater used for the wet wear test (row 93)  that does not seem compatible with the application target.

there are several papers that studied similar alloys e.g. "Batra, U., Batra, N., & Sharma, J. D. (2013). Wear performance of Cu-alloyed austempered ductile iron. Journal of materials engineering and performance, 22, 1136-1142.",  "Batra, U., Ray, S., & Prabhakar, S. R. (2004). Tensile properties of copper alloyed austempered ductile iron: Effect of austempering parameters. Journal of materials engineering and performance, 13, 537-541", "Boulifa, M. I., & Hadji, A. (2015). Effect of alloying elements on the mechanical behavior and wear of austempered ductile iron. Mechanics & Industry, 16(3), 304". Please highlight the overcoming of the state of the art of the proposed paper.

Reviewer 3 Report

Interesting paper on the wear behavior of ductile iron, concerning different austenizing temperatures. Overall, the paper is good and it fits the scope of this journal. Nonetheless, some amends are required,

English should be revised thoroughly.

Lines 87 to 89: What was the testing conditions used in hardness testing? Add standard deviation to table 2. The indentations might have been influenced by the large nodules.  Also, a deeper discussion on how the microstructure affects the hardness should also be added to the discussion.

Line 109: When mentioning Figures is should be: Figure XXX a), b) and c); not “Figure 3a,b,c”.

Line 216: Specify what is DI

Line 81: Is “[20]” the equation or the reference from where it comes from?. Please rearrange in an easier to read format by mentioning the equation in the text with the reference. Then show the equation separately. Same for the equation in Line 84(which should be Equation 2).

Line 66 why this temperature?

Fig 12: any EDS to confirm the composition?

Fig 13. The symbols from 300-wet and 360-wet are not easy to interpret. Try different symbols maybe with different sizes to ease the interpretation

Lines 358 to 269: The conclusion section should be improved. It should be mentioned is the longer Austenitizing temperature help wear resistance or not.

Reviewer 4 Report

Suggestions for authors

1.     It is recommended to improve the introduction session and also mention the novelty of the work.

2.     Why the work is important? Define the Research gap

3.     Need to refine the Figure 1 of the manuscript. It is necessary to add scale.

4.     What is the reason for choosing the represented temperatures of the study?

5.     It is recommended to mention the load applied during hardness measurement in session 2.4.

6.     What is the reason for choosing mentioned tribological input parameters in both dry and wet test? Mention it in the session 2.5.

7.     How the authors measure the austenite and carbon content using X-ray measurement? Although the XRD peaks looks similar in all cases. It is recommended to explain it.

8.     Justify “decrease in austenite leads to increase in hardness of the alloy”

9.     Same data is represented in the form of table as well figure (table 2 and figure 5 (a – b), figure 6). It is recommended to delete anyone of the representation.

10.  It is recommended to check the representation of wear rate. In general, wear rate is represented as (mm3/N.m or mg/N.m).

11.  It is recommended to explain more on session 3.3. What is the significance of figure 9 and Figure 12.

12.  It is recommended to modify the conclusion.

Reviewer 5 Report

Dear Editor-in-Chief 

Concerning my revision of the paper titled: Microstructure and dry/wet tribological behaviors of 1% Cu alloyed austempered ductile iron"

Manuscript ID: materials-2240855

I have gone through the article. I found that the authors investigated the various austempering conditions to 1 wt.% Cu alloyed ductile iron to obtain different austempered ductile irons (ADIs). They have done good scientific work with new information and provided enough experimental details to reproduce it. So I recommend its publication in the Materials Journal after minor revision by taking up the following corrections/suggestions:

1.    English language should be revised thoroughly.

2.    The authors are directed to include the motivation of the work.

3.    Should the authors include the work's real application(s)?

4.    More comparative studies should be provided in DISCUSSIONS to explore the importance of your work.

5.    The references need to be updated to 2022.

6. The resolution of Figs. 4,7, and 10 needs to be modified.

Round 2

Reviewer 1 Report

The article has been improved in the aspects indicated in the previous review report. It only remains for the kilo prefix to be corrected (k instead of K), on pages 2 and 3.

Author Response

Point 1: The article has been improved in the aspects indicated in the previous review report. It only remains for the kilo prefix to be corrected (k instead of K), on pages 2 and 3.

Response 1: Thanks to the reviewer's comment, the mentioned errors have been corrected in the revised manuscript (as the highlighted on pages 2 and 3).

Reviewer 2 Report

The authors integrated the paper and answered the reviewers' observations.

Author Response

Point 1: The authors integrated the paper and answered the reviewers' observations.

Response 1: Many thanks to the reviewer for giving the positive comment.

Reviewer 3 Report

The authors answered my questions and revised accordingly.  Still, the conclusion only presents topics. This section would benefit form a well-writen text. It can be accepted after the conclusions are presented in a clear and concise manner."

Author Response

Point 1: The authors answered my questions and revised accordingly.  Still, the conclusion only presents topics. This section would benefit form a well-written text. It can be accepted after the conclusions are presented in a clear and concise manner."

Response 1: As suggested, the conclusion has been modified to clearly present the main findings of this study in the revised manuscript (as the highlighted).

Reviewer 4 Report

Suggestions to authors:

1. Check the x axis values in Figure 1 (1,5,10 ?)

2. Application of dry sliding wear is not clear, need to describe it. 

3. It recommended to refine 3rd point of the conclusion. The word "best" is not recommended to use in research article. In general, the wear rate of less than 10^-6 should be used for engineering applications. 

4. Recommended to cite articles of research times. 

Author Response

Point 1: Check the x axis values in Figure 1 (1,5,10 ?)

Response 1: As suggested, Figure 1 has been reworked to represent the X-axis at 1,5,10 hours.

Point 2: Application of dry sliding wear is not clear, need to describe it.

Response 2: Information about ADI in dry sliding wear applications has been added to the revision. The content “ADI is often used in environments with dry wear behavior, such as automotive gear components.” has been written in the revised manuscript (as highlighted in line 42-43).

Point 3: It recommended to refine 3rd point of the conclusion. The word "best" is not recommended to use in research article. In general, the wear rate of less than 10^-6 should be used for engineering applications.

Response 3: As suggested, the 3rd point of the conclusion has been modified, and the word “best” has been replaced with “excellent”. In addition, the use of unit is mg/Nm for wear rate in this study, if the unit is changed to g/Nm, it will become a 10-7 grade, which would be less than 10-6 as mentioned.

Point 4: Recommended to cite articles of research times.

Response 4: It is well known that there are many academic literature around the world that cannot all be cited in a paper. However, the authors have made every effort to cite relevant literature for this paper as references. All the cited literature are presented in the manuscript as the required format.
